# A Systematic Review of Analytical and Modelling Tools to Assess Climate Change Impacts and Adaptation on Coffee Agrosystems

**Muhammad Faraz** [1], **Valentina Mereu** [2], **Donatella Spano** [1,2], **Antonio Trabucco** [1,2], **Serena Marras** [1,2] **and Daniel El Chami** [3,*]

1   Department of Agriculture Science, University of Sassari, Viale Italia 39/A, I-07100 Sassari, Italy
2   Euro-Mediterranean Centre on Climate Change (CMCC) Foundation, Impacts on Agriculture, Forests and Ecosystem Services (IAFES) Division, Via De Nicola 9, I-07100 Sassari, Italy
3   TIMAC AGRO Italia S.p.A., S.P.13, Località Ca' Nova, I-26010 Ripalta Arpina, Italy
*   Correspondence: daniel.elchami@roullier.com; Tel.: +39-0373-669-111

**Abstract:** Several modelling tools reported the climate change impact on the coffee agrosystems. This article has adopted a systematic approach to searching out information from the literature about different modelling approaches to assess climate change impacts or/and adaptation on coffee crops worldwide. The review included all scientific publications from the date of the first relevant article until the end of 2022 and screened 60 relevant articles. Most results report research conducted in America, followed by Africa. The models assessed in the literature generally incorporate Intergovernmental Panel on Climate Change (IPCC) emission scenarios (80% of manuscripts), particularly Representative Concentration Pathways (RCP) and Special Report on Emission Scenarios (SRES), with the most common projection periods until 2050 (50% of documents). The selected manuscripts contain qualitative and quantitative modelling tools to simulate climate impact on crop suitability (55% of results), crop productivity (25% of studies), and pests and diseases (20% of the results). According to the analysed literature, MaxEnt is the leading machine learning model to assess the climate suitability of coffee agrosystems. The most authentic and reliable model in pest distribution is the Insect Life Cycle Modelling Software (ILCYM) (version 4.0). Scientific evidence shows a lack of adaptation modelling, especially in shading and irrigation practices, which crop models can assess. Therefore, it is recommended to fill this scientific gap by generating modelling tools to understand better coffee crop phenology and its adaptation under different climate scenarios to support adaptation strategies in coffee-producing countries, especially for the Robusta coffee species, where a lack of studies is reported (6% of the results), even though this species represents 40% of the total coffee production.

**Keywords:** coffee agrosystems; climate change (CC); impact; adaptation; modelling; IPCC scenarios

## 1. Introduction

The last Intergovernmental Panel on Climate Change (IPCC) Report [1] gathered extensive evidence that climate change had caused substantial damage and increasingly irreversible losses over terrestrial and marine ecosystems and natural resources. Agriculture is among the sectors most affected by climate change, mainly due to extreme events' increased frequency and intensity, with worsening expectations [2].

Climate change is estimated to increase agricultural production and food access pressures, especially in vulnerable regions, thus undermining food security and human nutrition [1]. It has altered hydrological cycles; extreme events such as droughts, floods, storms, heat waves, and other abnormalities on Earth are becoming more common [3,4]. The uncertainty in precipitation patterns, more intense rainfall, and the increase in soil erosion are regarded as direct climate change impacts on the agrosystems [5,6], which generate abiotic stress on biodiversity. Indeed, flooding and surface runoff are vehicles

of soil nutrients, pesticides, and other harmful chemicals into freshwater, depleting soil fertility and polluting groundwater resources [7]. Water scarcity and temperature rise affect plants' biochemical and physiological processes [8]. Increasing temperatures have also caused a substantial decline in crop production and are considered a high risk for crops in the future [9,10], especially at mid and low latitudes. The impact of climate change on the productivity of several staple crops is foreseen to be critical in low-latitude tropical regions [11].

Coffee is one of the most important crops in low-latitude regions where climate changes are expected to impact agricultural systems heavily [2]. Thus, arable land in tropical and subtropical regions may lose a considerable amount of such areas by 2050; for example, South America may lose 1–21%, Africa 1–18%, Europe 11–17%, and India 2–4% [12]. Another study illustrated a critical level of water deficit (0.82 kPa) during the flowering stage of Arabica coffee, after which the yield significantly declined, and predicted that about 90% of countries will breach this benchmark if warming rises to 2.9 °C by 2095 [13].

Coffee is cultivated worldwide by about 20–25 million smallholder farmers on 11 million ha of arable land spread across 60 tropical regions [14]. The international trade in coffee commodities is ranked second after petroleum products. Developing countries contribute considerably to exports to more industrialised countries. The United States imports approximately 23% of total traded coffee beans, while the European Union imports about 43% [15]. The estimated coffee consumption is more than 400 billion cups per year, and almost 100 million people are engaged in this industry and derive their income directly or indirectly from coffee commodities [15,16].

The main commercial coffee species are Arabica (*Coffea arabica* L.) and Robusta (*Coffea canephora* L.), accounting for 99% of the total coffee production, where the individual share of both species is 60% and 40%, respectively [17]. Meanwhile, the worldwide prediction for the productivity of Arabica coffee is 35.5% higher than that of Robusta coffee. Brazil, Vietnam, Indonesia, and Colombia are the leading countries globally, contributing to 68% of the international market [15]. The production of the major coffee species and cumulative production by both species within the last five years, produced by the leading countries worldwide, is shown in Figure 1.

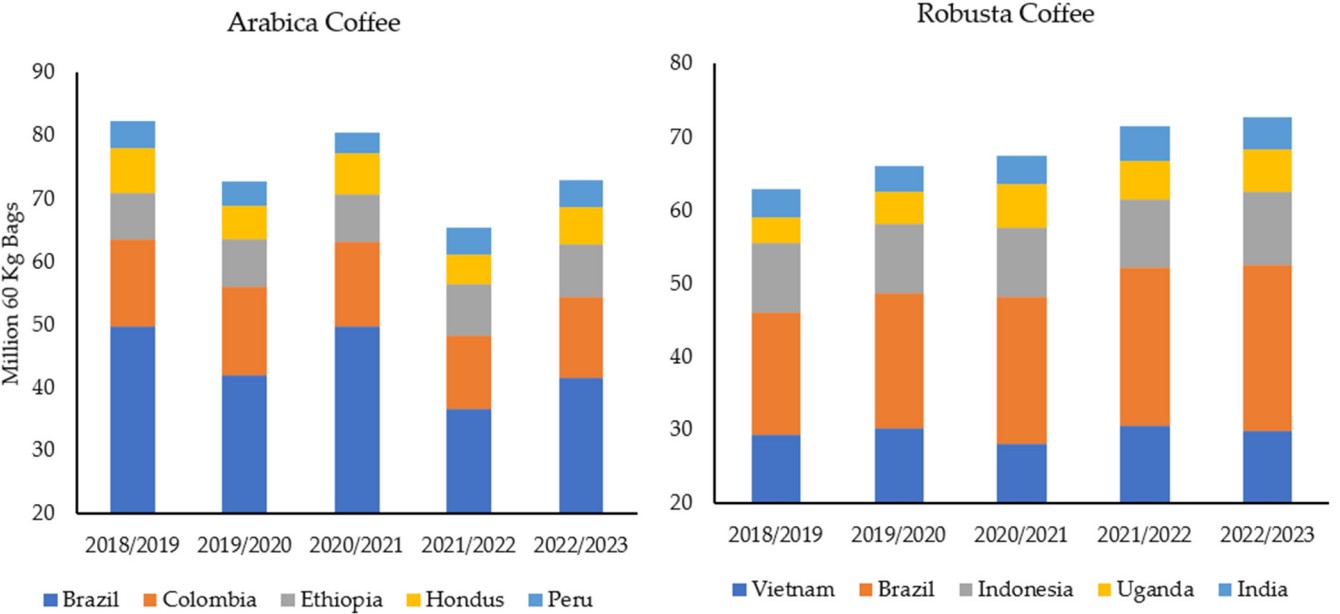

**Figure 1.** Coffee production in the top five countries worldwide for Arabica and Robusta coffee (data extracted from [18] for 2018–2022).

Climate changes affect coffee crop production due to more frequent insect and pest diseases induced by climate variability [19]. Moreover, high temperatures and reduced precipitation considerably affect the flowering and fruiting of coffee plants and the quality of the beans [20,21]. Coffee crops begin to bloom after the first spring rainfall, but under drought conditions, fewer flowers will sprout, consequently reducing fruit development. On the other hand, under heavy rain conditions, yield loss occurs as flowers and fruits fall off the tree [22,23]. The vegetative and reproductive phases are specifically dependent upon temperature. Temperature rise accelerates the berry ripening, reducing the bean filling duration. Low temperatures lead to defoliation and decreased photosynthesis, causing fluctuation in leaf mass [24]. In addition, low seasonal rainfall is causing branch death, reducing fruit setting resources and damaging coffee beans [25].

Arabica adapts better at high altitudes with an optimal temperature range of 18–22 °C; in contrast, Robusta thrives at lower altitudes with optimal temperatures between 22 and 28 °C. However, neither species can produce abundant yields under adverse conditions nor maintain beverage quality [26].

Several studies have modelled the impact of climate change on coffee production, depicting an upward shift of the crop along with yield losses at lower latitudes [27,28]. Climate uncertainty will affect the coffee production of 9.5 billion kg year$^{-1}$ obtained in 2018, with a 50% reduction by 2050 in suitable environments, thus putting a heavy toll on the coffee trade as a threefold rise in demand is also expected [29]. Specifically, global warming will significantly affect coffee crop production worldwide, with a reduction in 2050 of up to 60% in southern Brazil [30], 90% in Nicaragua [19], and 30–60% in Kenya [31]. Both Robusta and Arabica will be negatively affected by increasing temperature: a 1 °C increase in minimum/maximum temperature (16.2/24 °C) could result in ≈14% or 350–460 kg ha$^{-1}$ Robusta yield reduction [32], even though the Arabica favourable environment could be relocated to 300 m up the altitude gradient in Nicaragua [19]. In addition, high temperatures would make coffee farming susceptible to fungal attacks, such as coffee rust, at lower altitudes and borer damage at high elevations [33,34].

The available literature on coffee presents extensive insights and recommendations for using models and other analytical tools to study climate change impacts and adaptations in coffee production in different regions, such as [35] in Central America, [32] in Vietnam, [36] in Brazil, [37] in Colombia, [38] in Uganda, [39] in Ethiopia, and many more. While the impacts of climate change on coffee have been systematically studied [40], modelling tools still have not received enough attention in terms of systematic review and classification.

The current review is designed for a comprehensive view of the models and tools available to investigate the implications of climate change conditions on coffee growth and yield. The study will also help identify the potential gaps and future trends for research studies to improve modelling tools to guide farming towards sustainable and resilient management of coffee cultivation under climate change conditions. With this aim, a systematic review approach, already consolidated in the climate and agricultural sciences [41–43], is applied to explore the different modelling tools used to investigate climate change impacts and adaptation on the two major coffee species, Robusta and Arabica. We gave special attention to highlighting the eventual capacity of the available tools to assess the effectiveness of adaptation options.

## 2. Materials and Methods

The Collaboration for Environmental Evidence (CEE) described a systematic review guideline in which PECO or PICO elements demonstrate the research question in various components [44]. Based on a proper methodology, the research question was formulated, and proposed the following:

"What are the analytical tools for coffee crop modelling under climatic uncertainties?"

Based on this question, we developed the PICO elements and the search keywords in Table 1. Once created, we tested the keywords on different search engines, such as Web of Science, Scopus, and Science Direct, on 27 July 2021 (Table 2). To reduce incompatibilities

between various search engines, we avoided the excessive use of operators (e.g., wildcard, Boolean, braces, etc.). We extracted the complete database on 13 February 2023.

**Table 1.** The breakdown of the research question into PICO components and related keywords.

| PICO | Description | Keywords |
|---|---|---|
| Population | Coffee production, focusing on agrosystems and bean production but excluding the processing phases following post-harvest. The study includes impacts, adaptation, and resilience to all climate variables (temperature, rainfall, $CO_2$). | Coffee, crop, tree, production, agrosystems, farm. |
| Intervention | The intervention is the tools used to assess impacts, adaptation, and resilience to climate change—variability in temperature and precipitation. The review will consider no time scale. It will include all scenarios investigated in the literature. | Climate change, impact, adaptation, resilience, GHG emission, climate Variable. |
| Comparator | Qualitative vs. quantitative models; mathematical vs. biophysical models; spatial modelling. | |
| Outcome | Modelling techniques. Analytical tools. Programming. | Models, modelling, tools, programming. |

**Table 2.** Development, trial, refinement, and screening of search terms. The keywords in bold represent the selected ones since they show a reasonable hit in all databases.

| Search Term | Science Direct | WoS (All Fields) | Scopus (Title-Abs-Key) | Comments |
|---|---|---|---|---|
| "climate change" AND coffee | 5573 | 536 | 538 | The search term might include adaptation and resilience of coffee to climate change. It will also include other aspects related to impacts and mitigation or policy documents |
| "climate change" AND coffee AND (impact OR resilience OR adaptation) | 4948 | 327 | 299 | A good search term. A reasonable number of hits, which include all the words needed to answer the research question. |
| "climate change" AND coffee AND model AND (impact OR resilience OR adaptation) | 3946 | 113 | 89 | Somehow restrictive search term. |
| "climate change" AND coffee AND (model OR programme OR tool) | 5212 | 217 | 187 | A good search term. A reasonable number of hits which include all the words needed to answer the research question. |
| "climate change" AND coffee AND (model OR programme OR tool) AND (impact OR resilience OR adaptation) | 4707 | 149 | 118 | A good search term. A reasonable number of hits which include all the words needed to answer the research question. |
| **climate AND coffee AND (model OR programme OR tool)** | **15,161** | **387** | **381** | **A good search term. A reasonable number of hits which include all the words needed to answer the research question.** |
| climate AND coffee AND (model OR programme OR tool) AND (impact OR resilience OR adaptation) | 11,453 | 187 | 147 | A good search term. A reasonable number of hits which include all the words needed to answer the research question. |

Besides database sources, the systematic review used search engines and organisation websites in which a maximum of 50 'hits' were recorded from each website (Table 3).

**Table 3.** List of academic database sources and websites used.

| Database Sources | Search Websites | Organisation Websites |
|---|---|---|
| Web of Science (WoS) Scopus<br><br>Science Direct | google.com (accessed on 13 February 2023).<br>googlescholar.com (accessed on 13 February 2023). | World Bank<br>FAO<br>Consultative Group on International Agricultural Research (CGIAR)<br>International Fund for Agricultural Development (IFAD)<br>Natural Resources Institute<br>Climate Institute<br>Coffee & Climate<br>International Trade Centre<br>Fairtrade<br>Coffee Research Institute<br>International Coffee Organisation |

For the literature screening, we adopted the following inclusion criteria: (i) subject relevant (anywhere in the world, small landholder farmer or commercial system); (ii) type of intervention (climate scenario available in the literature, tools to assess impact resilience to climate change); (iii) comparator (Spatial modelling); (iv) method (Qualitative vs. quantitative modelling); (v) outcome (studies that consider production modelling).

The effect modifier restricted access to limited primary data, and less variability in modelling and potential impacts (GHG emission scenarios, crop varieties, different production systems and techniques, different agro-ecological conditions, etc.) was unavoidable. Therefore, the review team agreed to adopt narrative analysis and, where possible, quantitative evidence instead of meta-analysis. Interpreting broad subjects with a narrative approach is more suitable, producing a disparate range of outcomes. The narrative analysis approach can acquire the attention of stakeholders and decision-makers by providing them with research gaps in targeted research areas [40–43]. The review team carefully reduced any source of biases in evaluating climate change mitigation and adaptation impacts on the coffee cropping systems.

The literature review did not include a timeframe and was extended until 31 December 2022, based on different search keywords tested on 27 July 2021. Available literature published in English was considered, without specific field restrictions. Keyword search outcomes were recorded and exported to "Mendeley" (a bibliographic software package, 2.100.0). The inclusion criteria were applied by selecting relevant title papers, then abstract evaluation, and, finally, reviewing full texts (Figure 2). Obtained data were tabulated using a common spreadsheet format (i.e., MS Excel). During data extraction, transparency was ensured to avoid heterogeneity in data documentation, and all the review steps were recorded using the PRISMA checklist.

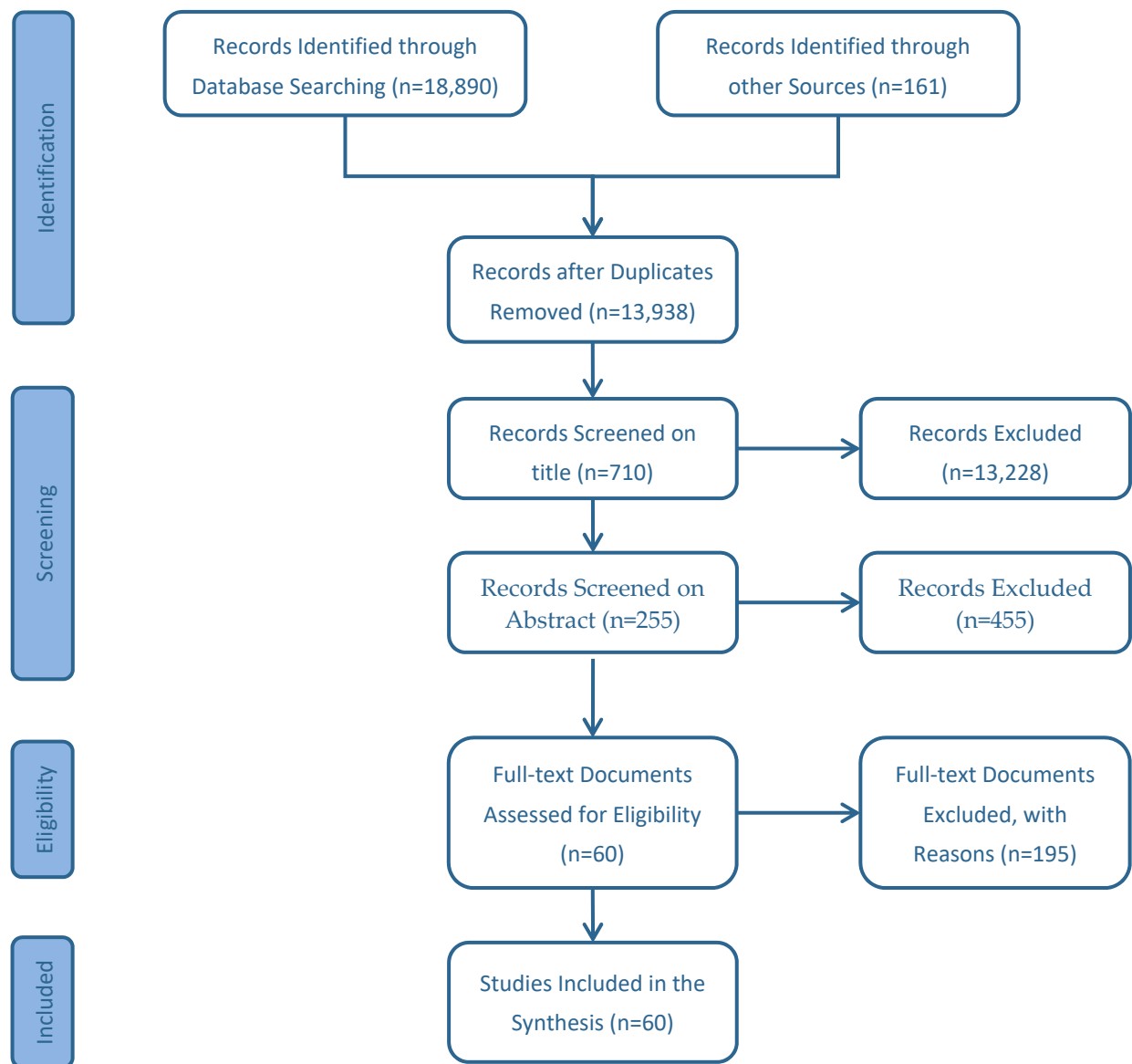

**Figure 2.** Flow diagram of the systematic review process (after [45]).

## 3. Results

The results are divided into three different sections. They will first present a general statistical analysis of the screened database, then assess the latter according to climate change processes reviewed to address the models and tools found in the literature in the final section.

### 3.1. Bibliometric Analysis

A total of 60 eligible studies were retrieved from the literature on modelling climate-driven aspects related to coffee production (Figure 2). The documents were comprehensively searched and categorised according to different categories: region, year of publication, model type, data used to validate the models, coffee species, climate scenarios, climate impact, and climate adaptation (Supplementary Materials). All documents showing simulation models and producing predictions for specific periods for data synthesis were considered. The number of documents were identified and classified according to publication years. Indeed, even though not constantly linear, the trend over time in publication numbers shows an increase over the last decade (Figure 3), with peaks in the number of

publications in 2017 and 2022 (*n* = 8, and *n* = 9), while also 2011, 2015, 2018, and 2020 show a consistent number of publications (i.e., 5 and 6).

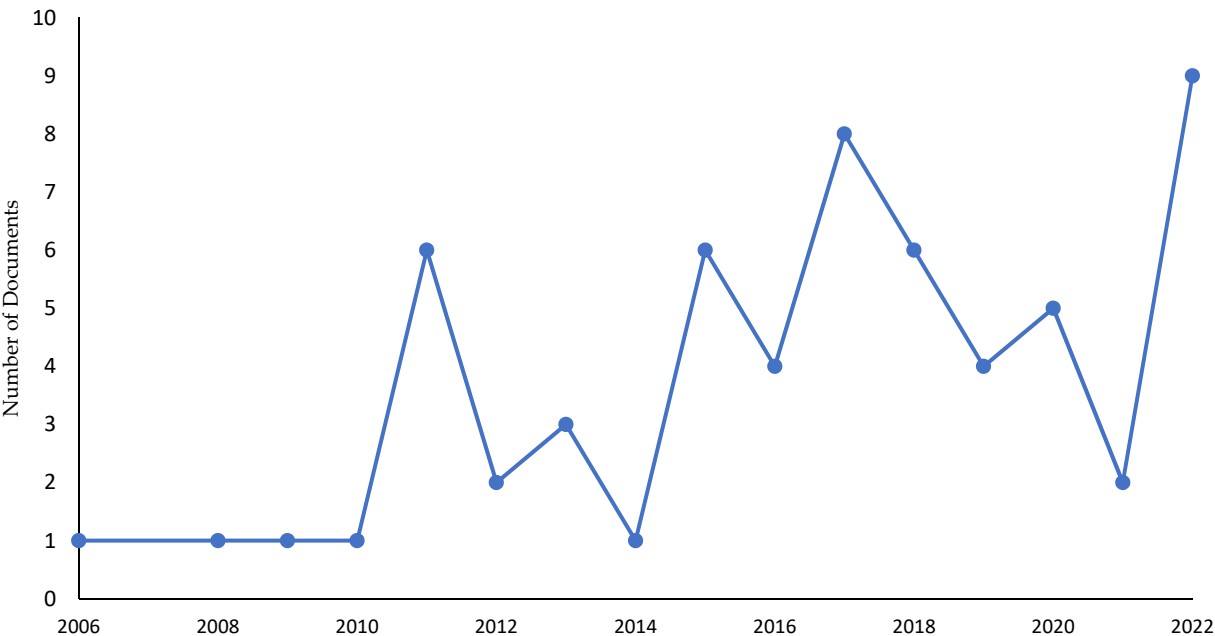

**Figure 3.** Documents published each year depicting modelling tools to simulate climate change impacts and adaptation on coffee production until 2022.

From another side, Figure 4 depicts the literature published on coffee production in different countries, most of which relate to Brazil (*n* = 14). The research on the coffee crop in Ethiopia is reported in seven documents, whereas we obtained a similar number of documents (*n* = 2) for Indonesia, Tanzania, Mexico, Colombia, Zimbabwe, and Costa Rica. Some articles combined studies including many countries (*n* = 14), and a few (*n* = 4) analysed climate-related aspects of coffee crops worldwide.

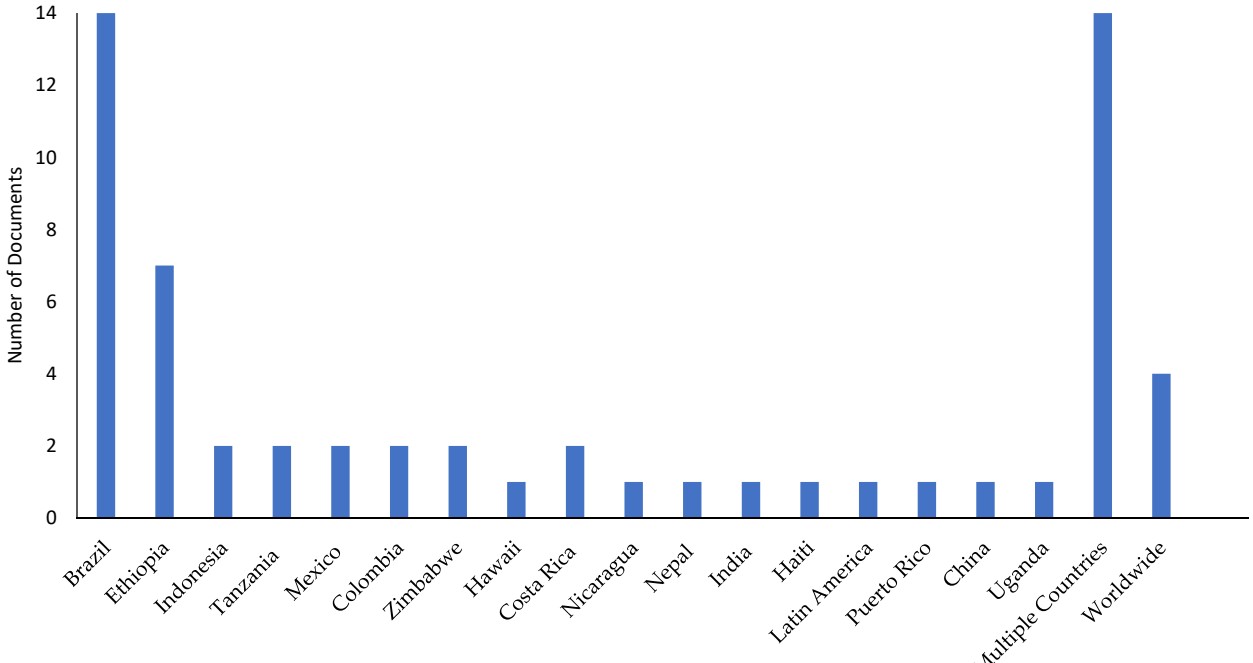

**Figure 4.** The number of documents published in different countries modelling impacts and adaptation to climate change in coffee production till 2022.

The available literature mainly focused on *Coffea arabica* L., as reported in Table 4. Most documents included results for the American (South America = 18, North America = 15) and the African (*n* = 20) continents and four papers carried out global research, focusing on *Coffea arabica* L. species. A few documents did not define the assessed species (*n* = 6). Among document types, the available literature is covered mainly by research articles (*n* = 55).

**Table 4.** The number of documents published in different continents, document types, and coffee species.

| Continents | Document Type | | | Species | | | | Total |
|---|---|---|---|---|---|---|---|---|
| | **Research** | **Chapter** | **Report** | *Coffee arabica* **L.** | *Coffea robusta* **L.** | **Both Species** | **Not Mentioned** | |
| **North America** | 12 | 1 | 2 | 12 | | 1 | 2 | **15** |
| **South America** | 18 | | | 15 | 1 | 1 | 1 | **18** |
| **Africa** | 18 | 1 | 1 | 15 | 2 | 1 | 2 | **20** |
| **Asia** | 3 | | | 1 | | 1 | 1 | **3** |
| **Worldwide** | 4 | | | 3 | | 1 | | **4** |
| **Total** | **55** | **2** | **3** | **46** | **3** | **5** | **6** | **60** |

### 3.2. Processes Reviewed

The results identified two climate change processes using modelling tools: impacts and adaptation. The values in Figure 5 refer to the percentage of documents related to climate variability's effect on coffee production.

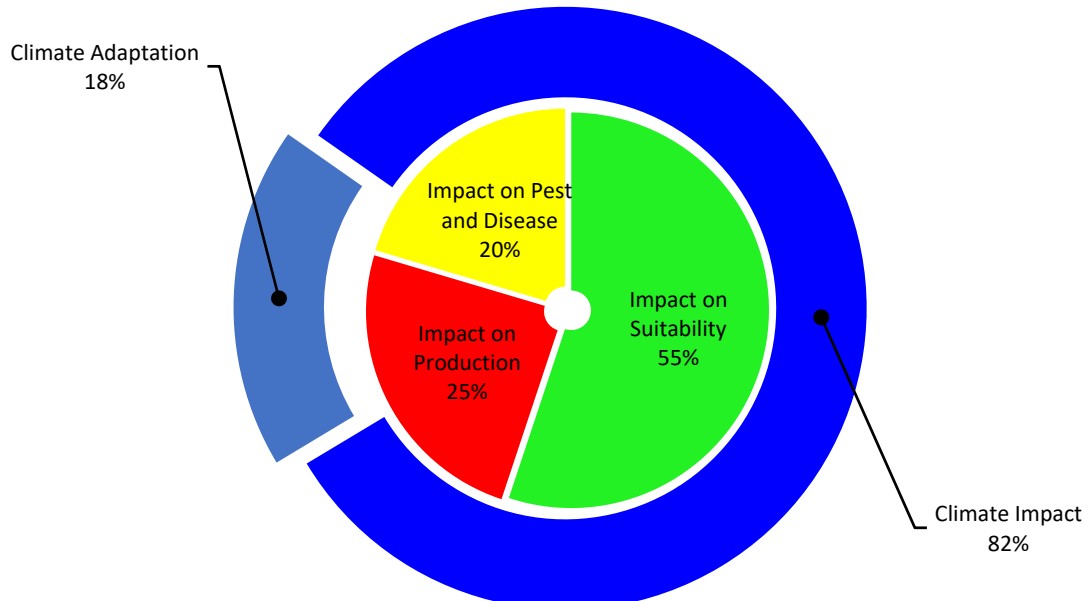

**Figure 5.** Percentage of documents reviewed according to climate process (climate impact, adaptation). The climate impact is further divided into three categories: impact on suitability, impact on production, and impact on pests and diseases.

The impacts of climate change on coffee production are thoroughly assessed in the literature (82%, in particular, the climate suitability (55%), increased incidence of pests and disease (20%), and decline in production (25%). A consistent number of reviewed studies proposed different adaptation strategies (18%).

### 3.3. Analysis of Models and Tools

A model is a simplified representation of reality though a functional scheme that allows one to investigate the properties of a system and, in some cases, predict its future outcome. Different models were developed for coffee crops to estimate current and future

production and distribution, considering climate variability as a driving factor. Based on the review, the models were classified into deterministic, stochastic, and mixed stochastic/deterministic (Table 5). Deterministic models do not account for randomness in data, nor have a probability function, so a set of inputs and established relationships determines the output.

**Table 5.** Classification of the reviewed models.

| | | Model Types | | | Total |
|---|---|---|---|---|---|
| | | **Deterministic Models** | **Stochastic Models** | **Deterministic and Stochastic** | |
| Model Category | Regression models | 12 | 3 | | **15** |
| | Crop models | 2 | 8 | | **10** |
| | Species distribution models | 3 | 30 | 1 | **34** |
| **Total Models** | | **17** | **41** | **1** | **59** |

On the contrary, a stochastic model includes a random component that uses a distribution as one of the inputs and results in a distribution as output. It presents data and predicts outcomes that account for certain levels of unpredictability or randomness. In addition, some models are stochastic and become deterministic after training. The training installs rules into a network that prescribes its behaviours, so an untrained model shows inconsistent behaviours. These models were included in the mixed model type.

The Eta model was excluded from this classification; it was used once in the literature to assess future suitability ranges, expressed in percentage, of coffee in Southeast Brazil based on annual mean water and temperature restrictions of the Arabica coffee [46].

Among the other 59 models identified in the literature, as reported in Table 5, 41 are stochastic models. Species distribution models are the most common (*n* = 34), followed by crop models (*n* = 10). Additionally, species distribution models include deterministic and stochastic models. Hence, the total number of deterministic models is 17, most of them being regression models (*n* = 12).

### 3.3.1. Models' Categories

In addition to the first model's classification, another conceptual modelling classification was applied to the modelling tools based on the type of mathematical function/process used to estimate climate change impacts and adaptation. This classification identified regression, crop, and species distribution models as the three main categories (Table 5). A description of each model category is reported in the following sections.

#### Regression Models

Regression models are simple models used to establish relationships between climate (and other environmental) variables and crop outcomes by fitting regression equations. These models are effective in detecting more general trends and projecting future scenarios. The literature comprises various regression models, including non-linear regression models, multivariate analysis, AutoRegressive Integrated Moving Average (ARIMA) models, climate-based statistical models, econometric models, Generalised Linear Model (GLM), and the Generalised Additive Model (GAM).

Regression models are extensively used to study the impact of climate change on coffee crop yields. Climate-based statistical models, such as those used by [22], predict yields of *Coffea arabica* L. and *Coffea robusta* L. in India for specific years (2010–2012) based on temperature, rainfall, and humidity variables. ARIMA models are employed by [47,48] to assess the influence of climate change on coffee yield in Tanzania and Brazil. The linear regression models predict coffee yield in Ethiopia until 2060 and investigate climate suitability for Arabica coffee until 2080 [49,50].

Non-linear regression equations are applied to study the effects of factors, such as temperature, leaf wetness, and distribution in the sun and shade-grown systems, on coffee rust and coffee crop vulnerability in Brazil and provide projections under climate change until 2080 [51–54]. Bio-economic models are used to predict the impact of the Coffee Berry Borer (CBB) on coffee crops in East Africa and examine the influence of climate on CBB in both full-sun and shade-grown systems [55]. Various studies used econometric models to investigate the correlation between coffee yield and climate variables and identify climate-vulnerable areas in different countries, such as Mexico, Brazil, and Colombia. The principal component analysis is also used to predict the vulnerability in Brazil's coffee region to climate change until 2080 [20,28,56].

Crop Process-Based Models

Crop models are process-based models that simulate the growth and development of crops in specific environmental conditions, and simulate biogeochemical processes to predict crop growth and yields and optimise crop management strategies under present and projected climatic conditions. However, these models require extensive effort in equations and parameter calibration. In the existing literature, several crop models have been identified, including mechanistic models, the yield-safe model, the DynaCof model, dynamic models, Irrigation Management System (IManSys) model, and agrometeorological models.

Several studies applying crop models (e.g., DynaCof) focus explicitly on agroforestry systems and compare them with open sun-grown systems under changing climate variables in Costa Rica, Guatemala, Nicaragua, Colombia, and Brazil [24,57–59]. The yield-safe model determines coffee yield under changing climate scenarios in Ethiopia [21]. Agrometeorological models incorporate irrigation methods to counter the effects of high temperatures and frost from 2040 to 2070 in Brazil. Other studies focus on shade levels to mitigate drought intensity in East Africa, and the IManSys model is used to calculate irrigation requirements for coffee crops under IPCC scenarios in Hawaii [60–64].

Species Distribution Models

Species distribution models identify the distribution among environmental and spatial gradients of a particular species and confirm the suitability of its niche, considering climate impact and other environmental variables. Some reviewed models driven by machine-learning algorithms [39] can investigate climate suitability and include Maximum Entropy (MaxEnt), Random Forests (RF), Boosted Regression Trees (BRT), Generalised Boosted regression Model (GBM), Support Vector Machine (SVM), and Multivariate Adaptive Regression Spline (MARS) models. Moreover, other suitability models are (i) the agro-ecological land elevation model for *Coffea arabica* L. (ALIECA), (ii) the EcoCrop Model, and (iii) the crop niche selection for tropical agriculture (CaNaStA).

Furthermore, the ensemble modelling approach, which combines different models to perform specific scientific activities, has become more common lately to ensure the projections' reliability and reduce modelling uncertainty. Other various modelling tools exist for pest species distribution and disease occurrence. They are (i) the bio-economic models, (ii) the empirical disease models, (iii) the Dinamica EGO model, (iv) the Climex model, (v) the thermal constant model, and (vi) the Insect Life Cycle Modelling Software (ILCYM). The species distribution models are more common in the literature.

The MaxEnt model is widely used to assess the climate suitability of Arabica coffee and predict its future implications worldwide. The model uses various environmental factors as explanatory variables, including temperature, precipitation, aridity, evapotranspiration, soil slope, and land cover [65], to simulate (i) climate suitability in Nepal, Indonesia and Haiti, (ii) the impact on indigenous Arabica coffee in Sudan and Ethiopia, (iii) the adaptation strategy for coffee communities in Mexico, and (iv) to assess climate vulnerability in Puerto Rico by 2099 and Mesoamerica by 2050 [66–70].

Furthermore, the MaxEnt model applied in Indonesia and Zimbabwe produced climate suitability until 2050 and extended projections for China (2060) and Ethiopia

(2070) [71–74]. It also assesses the coffee-pollinating species occurrence in Latin America in response to climate variability (temperature, precipitation, and dry season) and sets the suitable ecological zones in Costa Rica against temperature, elevation, and diurnal range [75,76]. Finally, the MaxEnt model assesses agroforestry systems for adaptation in Brazil and Mesoamerica by 2050, respectively, in response to temperature, precipitation, and bioclimatic variables [30,77].

The influence of climate variability (temperature, rainfall, and evapotranspiration) on Arabica agroclimatic zoning and coffee production was also investigated in Brazil using Eta, a regional climate modelling tool [46].

Random Forest (RF) models are run worldwide to classify the agro-ecological zones for Arabica coffee based on climate variables (temperature, precipitation, and dry months) [78]. An empirical disease model determined the incubation period of coffee rust (*Hemileia vastatrix*) in response to maximum and minimum temperatures and interpolated them to make predictions in Brazil [79]. Ecological modelling tools are also used to evaluate Brazil's phoma leaf spot distribution related to temperature and relative humidity [80]. The Dinamica EGO model produces the distribution of understorey coffee occurrences in Ethiopia [81]. Generalised Regression Models (GRM) are applied globally to assess the impact of Vapour Pressure Development (VPD) on Arabica coffee yield. In Ethiopia, GRM evaluated the influences of extreme agroclimatic indicators on Arabica coffee's overground biomass (AGB) until 2060 [13,82].

In Zimbabwe, the Coffee White Borer (CWB) occurrence probability until 2050 is assessed against temperature and precipitation factors by an ensemble of modelling approaches (BRT and GLM models) [34]. Another ensemble approach uses several machine-learning algorithms (SVM, MaxEnt, and RF) to investigate the worldwide distribution of coffee crops (Arabica and Robusta coffee) [27]. Another ensemble of modelling techniques (GLM, MaxEnt, RF, MARS, GAM, and GBM) examines the resilience potential for Arabica coffee in Ethiopia and the risk extinction of wild Arabica species in Ethiopian and Sudan while taking into account several climate variables [83,84].

The MaxEnt and CaNaStA models also use climate variables, such as temperature and precipitation, as an ensemble of models to generate climate suitability and the quality of Arabica coffee in Nicaragua and evaluate adaptation and mitigation options in Central America [19,85]. Moreover, an integrated approach of machine-learning algorithms (BFT, RF, and SVM) investigate the influence of climate variability (temperature and precipitation) and topological (elevation, soil slope angle) and soil characteristics (pH, soil Cation Exchange Capacity (CEC), apparent Bulk Density (BD), Soil Organic Carbon (SOC)) on the speciality of the coffee sector in Ethiopia under current and future scenarios [39]. Species distribution models (GAM, MaxEnt, and BRT) also predict Robusta's ecological and genomic vulnerability in its native region by 2050 [86].

The ALIECA model predicts the land suitability of Arabica coffee production using agro-ecological variables in Central America. An EcoCrop model assesses the climate suitability of a coffee-based cropping system in Uganda for the long term (2038) [87,88]. The Climex model accounts for the spatial distribution of CBB, considering the effect of environmental variables (temperature, moisture parameters, and other environmental constraints) [89]. Based on Brazil's air and soil temperature, the thermal constant model simulates the geographic distribution of coffee nematodes and leaf miners until 2080 [90]. Finally, the Insect Life Cycle Modelling software (ILCYM) predicts the coffee stink bug (*Antestiopsis thunbergii*) in Tanzania based on the Establishment of Risk Index (ERI), the Generation Index (GI), and the Activity Risk (AI) that corresponds to changes in critical factors/thresholds linked to coffee stink bug distribution based on air temperature [91].

### 3.3.2. Climate Change Scenarios

A total of 48 papers analyse the impact of future climate change conditions. The analyses follow the Intergovernmental Panel on Climate Change (IPCC) climate scenarios based on the emission and concentration of greenhouse gases in the atmosphere. Over

the years, the IPCC has developed several scenarios, including the Special Report on Emission Scenarios (SRES) for the third and fourth Assessment Reports (AR3 in 2001 and AR4 in 2007) [92], characterised by four qualitative storyline scenarios (A1, A2, B1, and B2) representing different demographic, social, and technological advancements. Further, the IPCC has recently developed advanced sets of scenarios: the Representative Concentration Pathway (RCP) and the Shared Socioeconomic Pathway (SSP), presented in AR4 [93] and AR6 [94], respectively. Both have a valid framework for projections until the end of the current century (2100). However, the RCP provides the concentration of greenhouse gases and radiative forcing levels associated with different emission pathways. In contrast, the SSP setup has a different approach considering greenhouse gas emissions, including population growth, economic development, energy use, land use, and other factors. The RCP is further subdivided into RCP 2.6 (optimistic), RCP 4.5 and 6.0 (intermediate), and RCP 8.5 (business as usual), and the set of the SSP scenarios are SSP126 (sustainable development), SSP245 (middle of the road development), SSP370 (regional rivalry), SSP460 (inequity), and SSP585 (full fossil-fuelled development) pathways.

The available literature has been more consolidated over the last decade. Therefore, the RCP are more prevalent than the SRES scenarios, commonly used by the literature in the earlier years (2008–2019), and were gradually, but not entirely, replaced by the RCP scenarios between 2015 and 2022. In the review results, the RCP scenarios prevail (23 papers), followed by the SRES (21 manuscripts) and the SSP (4 studies) (Figure 6A). The use and frequency of different scenarios across the results indicate that the highest number of papers (*n* = 7) adopted RCP 4.5 and RCP 8.5, followed by RCP 4.5, SRES A2A, SRES A2, and SRES A2 (*n* = 5 for each scenario). The other scenarios found in the literature (RCP (2.6, 4.5, 6.0 and 8.5), SRES (A2 and B2), and SSP (126, 245, 370 and 585)) are adopted in two research papers each (Figure 6B). The remaining scenarios were used only once in the literature, including RCP (2.6 and 8.5), (2.6 and 6.0), (2.6, 4.5 and 6.0), (2.6, 6.0 and 8.5), SRES (B1 and A2), (B1, and A1F1), (A2A, and A1F1), (A1B, A2A and B2A), (A2, B2, A2A and B2A), (A1, A1B and B2), and SSP (126, 245 and 585) (Figure 6C,D).

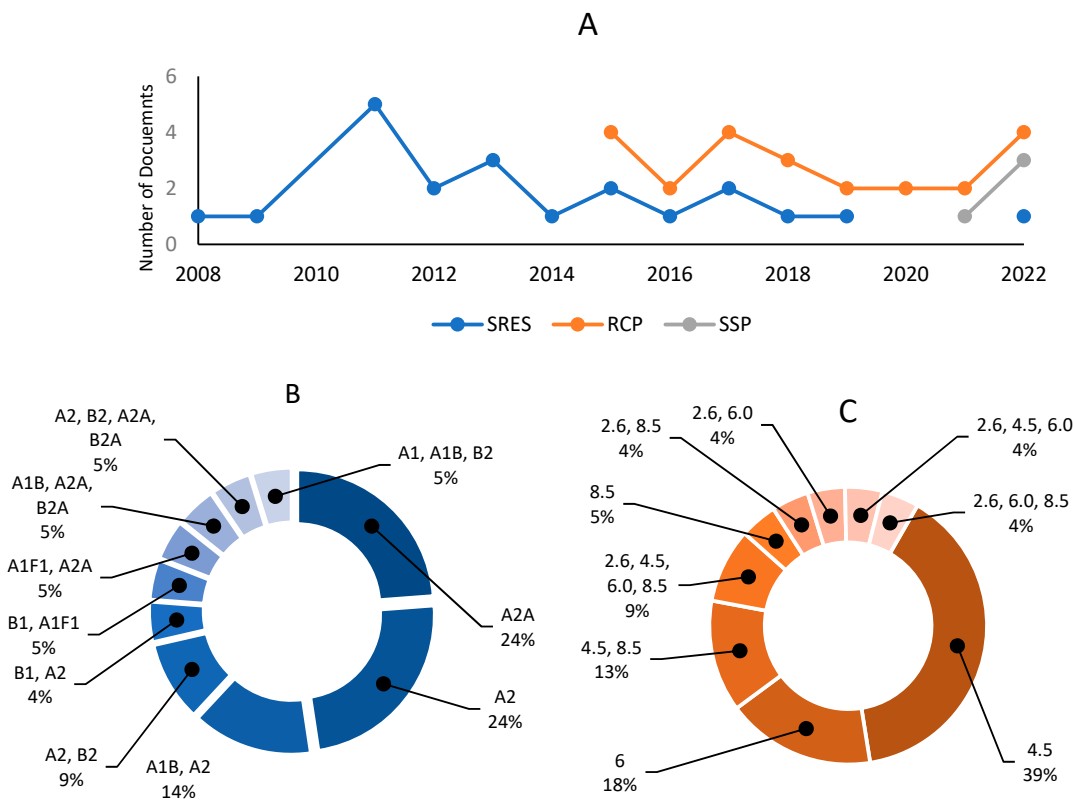

**Figure 6.** *Cont.*

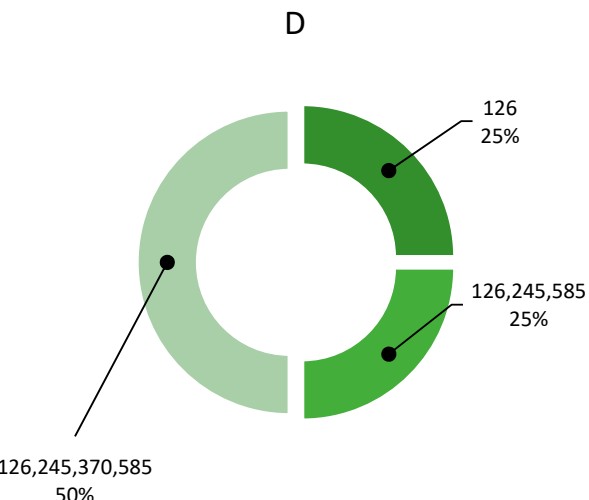

**Figure 6.** The coffee agrosystems are analysed using IPCC scenarios illustrating the trend of various scenarios (**A**), along with the frequency of sub-scenarios found in the literature, which are presented as SRES (**B**), RCPs (**C**), and SSPs (**D**), respectively.

## 4. Discussions

The literature was extensively searched to put together information on published modelling tools used to predict the impacts and adaptation of coffee agrosystems to climate change. Arabica is at a greater risk between the two major coffee species than Robusta; thus, research focused primarily on this species. Most models in the screened literature have incorporated the IPCC scenarios and evaluated climate change's impact on coffee agrosystems. Less attention is given to the adoption of adaptation practices. Most models have incorporated the IPCC scenarios, using projections until 2050, particularly the medium-emission sub-scenario RCP 4.5, either solely or combined with other sub-scenarios. The SERS' high-emission scenarios (A2) were also commonly used in the reviewed literature. The Supplementary Materials contain detailed information about each document screened for review.

Among the various regression models, econometric models based on multiple regression equations, which integrate the economic and climate variables, are commonly used to capture the effect of extreme events on coffee yield. The quadratic functional farm generates multi-collinearity, which does not affect the model's prediction but makes the estimator less accurate [28]. The AutoRegressive Integrated Moving Average (ARIMA) is particularly efficient in forecasting time series analysis, but its application with non-linear regression can compromise the model's accuracy [88]. The climate-based regression models monitor the coffee crop under critical stages, considering the effect of climate variables to analyse the coffee growth and developments at different growth cycles [22].

The MaxENT model is widely applied to determine the climate suitability of coffee crops. The model output is 1, considered the maximum probability, and 0, where species have a less suitable climate. The model calculates these values by dividing each weighted variable's sum by a scaling constant. This model is robust because it incorporates statistical and machine-learning techniques. However, the parameter selection is crucial, otherwise, the results may be biased. There is an inbuilt option to check the quality of the model using the Area Under Curve (AUC) index, as it provides a single overall measure of model accuracy [19,30].

A Random Forest (RF) model is also a popular machine learning classifier with high efficiency over large datasets without overfitting [22]. The Crop Niche Selection in Tropical Agriculture (CaNaSTA) is built on Bayesian statistics, aiming to determine a species' presence or absence and appraise the crop's performance. However, this model only works with its specific dataset format, and expertise in Bayesian statistics is also required, making it complicated and time-consuming.

Another model, EcoCrop (EC), determines the crop niche using environmental ranges, expressed in percentage, producing overall crop suitability [61]. The model gives individual suitability values for temperature and precipitation. Expert knowledge is essential for the model's accuracy in setting the crop parameters. The EcoCrop (EC) model can assess climate suitability even with limited ecological and environmental information [80].

The ensemble modelling approach is more efficient in suitability-related tasks because different machine learning and regression models perform together, highlighting modelling uncertainty and conservative choices for specific tasks. For example, integrating RF, CaNaStA, and MaxEnt has more precise results than the output produced by individual models [29]. The Dinamica's EGO model applies Weight of Evidence (WoE) to find the coffee occurrence understorey [74]. The agro-ecological land elevation model for *Coffea arabica* L. (ALIECA) is based on a Bayesian algorithm and provides information about land suitability in percentages, but does not provide data about the presence or absence of coffee crops. This model can also provide accurate results when the data are missing or uncertain [81].

The literature has applied and described several models for Identifying pests in coffee agroecosystems under climate change scenarios, including the ILCYM, climax, and thermal constant models. All these models have their strengths, but the ILCYM model stands out due to its ability to provide detailed information at a very high geographical scale, resulting in more precise results than other pest distribution models. However, the thermal constant model and ILCYM are based on temperature variables. They lack flexibility in accepting other climate variables important for pest–crop interactions, such as rainfall and relative humidity. Furthermore, neither model offers any crop or pest adaptation options. In addition, these models require daily or hourly data on a short time scale because pest–crop interactions may occur within 24 h [83,84]. The empirical disease and non-linear regression models use air temperature to determine the occurrence and intensity of diseases [72,89]. On the other hand, bioeconomic models are more flexible in considering shading, coffee berry borer (CBB) infestation, and temperature to generate information along with economic variables to estimate the shading value according to the disease infestation intensity [48].

Crop models are commonly used to assess climate impacts against adaptation strategies while modifying crop management systems. The available crop models recommend the shading level for optimal coffee yield in various regions [19,51,52]. However, few studies explored other adaptation options. As agrometeorological models, the Irrigation Management System (IManSys) model and the Yield SAFE model have been developed to enhance irrigation techniques and the efficiency of $CO_2$ fertilisation in a coffee production system [53,55,57]. All models obtained in each article are available in a supplementary file submitted with this manuscript.

## 5. Conclusions

Various modelling approaches have been applied to determine the climate change impact and adaptation of coffee agrosystems. The research adopted the systematic review approach to assess and classify the available literature according to (i) categories (regression models, crop models, species distribution models), (ii) types (deterministic, stochastic, mixed), and (iii) processes (climate impacts, climate adaptation). The results also included an assessment of the scenarios used to run different modelling tools.

In conclusion, machine learning models have complex algorithms and are stochastic, which produce predictions in situations where data include uncertainty or randomness, thereby generating more accurate results. The ILCYM model is particularly efficient in pest distribution due to its flexibility in accepting multiple variables, thus providing reliable data. The application of crop models was limited to a few studies on crop agrosystems.

Therefore, based on our results, we recommend intensifying adaptation research to explore the best options for different case studies. We advocate applying crop models to fill the gap for coffee phenology and propose adaptation strategies, for example, in-

troducing new varieties, water conservation methods, shading management at various altitudes, and soil organic matter management. The Robusta coffee species also needs to be further investigated in the literature because it is underestimated, despite having a higher adaptation potential.

We finally address these results to decision-makers to support scientific and applied policy design and implementation in climate change resilience and adaptation.

**Supplementary Materials:** The following supporting information can be downloaded at: https://www.mdpi.com/article/10.3390/su151914582/s1.

**Author Contributions:** Data curation, M.F. and D.E.C.; Formal analysis, M.F. and D.E.C.; Investigation, M.F. and D.E.C.; Methodology, M.F. and D.E.C.; Writing—original draft, M.F. and D.E.C.; Writing—review & editing, M.F. and D.E.C., V.M., A.T., S.M. and D.S.; Supervision, V.M., A.T., S.M. and D.S.; Funding acquisition, D.S. All authors have read and agreed to the published version of the manuscript.

**Funding:** This work was developed within the framework of the strategic project "CAT4—Agriculture Management Analysis for Adaptation" promoted by the Euro-Mediterranean Centre on Climate Change (CMCC) Foundation.

**Institutional Review Board Statement:** Not applicable.

**Informed Consent Statement:** Not applicable.

**Data Availability Statement:** Not applicable.

**Conflicts of Interest:** The authors declare no conflict of interest

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
