# Peer review of "A Systematic Review of Analytical and Modelling Tools to Assess Climate Change Impacts and Adaptation on Coffee Agrosystems"

_sustainability, doi:10.3390/su151914582_

Round 1

Reviewer 1 Report

The publication - A Systematic Review of Analytical and Modelling Tools to Assess Climate Change Impacts and Adaptation on Coffee Agrosystems - deals with a highly interesting and current topic.

It is a review study that examined 60 relevant articles. But although it is a very interesting topic, the authors did not make Chapter 3 (Results) very clear for the reader. It can be said that it is even opaque, and one gets lost in it very quickly. I would recommend putting some information in tables and adding only brief comments. This way, it is too difficult and unclear for the reader.

At the same time, for a review study, there are only 98 literature sources listed in the article, I would have expected more.

I also feel that the citations in the text of the paper are not formally correct - in accordance with the guidelines for authors. The page and line numbers are also not correct.

For Figure 1, I would recommend choosing a different type of graph, as it is not very clear and understandable. Similarly, Figure 5 is again unclear to me, and I would recommend trying to choose another more appropriate chart type.

The Discussion chapter is not a discussion in the true sense of the word, and I would rather call it a summary. Perhaps a table summarizing which study used which models, etc. would be useful here.

Author Response

We mean to express our gratitude for considering our manuscript for publication in ”Sustainability”.

Additionally, we would like to express our gratitude to the Reviewers for their insightful recommendations,

and remarks, which significantly aided in the improvement of the quality of our work.

We addressed all the comments raised by the reviewers, one by one. Either by changing the text accordingly and accepting all requests or by justifying whenever an explanation was required.

Please find the attached file where we have adressed the comments to the reviewer. 

Reviewer 2 Report

The manuscript with the title “A Systematic Review of Analytical and Modelling Tools to Assess Climate Change Impacts and Adaptation on Coffee Agro-systems”, is highly interesting dealing with one of the most beloved crops by billions of people – coffee. The manuscript revolves around current and potential challenges related to performance and sustainability of coffee crops under climate change effects.

Introduction - Very well written.

Material and Method section could be condensed a bit. Maybe tables 2 and 3 could be in supplementary files? Or write with smaller size font and put all 3 tables from material and method in on one single page. They are too spread out.

Results figure 2 – I consider still belongs to Material and Method. Only actual results shall be in Results section.

Overall, the results are a bit dissatisfying, because are predominantly an accounting of sources used and methods (approaches used in those papers, papers by country, percentage of papers that deal X subject etc). They are interesting but they look too much like some fact sheets available on the web, while this is a scientific journal. Authors should go more in depth at Results section with their scientific scrutiny and connect the results better with original discussion. Actually, I think Results and Discussion could be merged in this case. E.g. Lines 189-209 showcase very well what previous models have put into evidence. However, authors have to bring more of their original contribution to the discussion by using their critical insight more freely. What they have to say about the models used and their findings? Are they complementary or are the results contrasting/sufficiently reconcilable to enable an approach/consensus to be found?

Paragraph 320-326, is summarizing the overview findings. It is too simplistic – shading, irrigation and fertilization - far from touching the possibility of approaches. For example, crop improvement programs (creating varieties best adapted for given conditions) should be strongly considered, among others. ...

Authors have to examine a little bit the template of the journal and make sure they respect the editing format (reference style, reference list font, etc.).

Best regards.

syntax

Author Response

(The authors gave the same response as above.)

Reviewer 3 Report

General comments:

Title, before to read the manuscript, title seems apropriate, notwistanding it is not completely concording with the content of the manuscript. Systematic review, . . . for what?

On the other hand, at the end of manuscript, auhthorcal, ors are doing "recomending" technical recomendations, and, in opinion of this reviewer, discussion and conclussion must be based in the research. In this case, you are reviewing about the models used in the  coffe crop production by climate change issues. but you do not use the model, do not assess the climate change, etc

In fact, your OBJECTIVE is unclear, suggestion to be concise

Introduction; I think is good, however you must define the objective. Please revise your edition rules and I did not find some cites

M&M: respect the reviewed? or you are doing some simulations?

Of course, the "state of the art" in all subjects is always useful. Tip, highlight your review and maybe highlight the importance of using models in coffe under climate change conditios

others specific comments aare in the manuscript

Author Response

(The authors gave the same response as above.)

Round 2

Reviewer 2 Report

Dear authors,

the manuscript was significantly improved. All my concerns were addressed.

Best regards.